**Data Availability Statement:** All data regarding the temporal distribution of demand for COVID-19 related chest imaging at our institution during the

# Time series analysis of the demand for COVID-19 related chest imaging during the first wave of the SARS-CoV-2 pandemic: An explorative study

Daniel Koehler[1]*, Ann-Kathrin Ozga[2], Isabel Molwitz[1], Philipp May[1], Hanna Maria Görich[1], Sarah Keller[3], Gerhard Adam[1], Jin Yamamura[1]

1 Department of Diagnostic and Interventional Radiology and Nuclear Medicine, University Medical Center Hamburg-Eppendorf, Hamburg, Germany, 2 Institute of Medical Biometry and Epidemiology, University Medical Center Hamburg-Eppendorf, Hamburg, Germany, 3 Department of Radiology, Charité-Universitätsmedizin Berlin, corporate member of Freie Universität Berlin, Humboldt-Universität zu Berlin, and Berlin Institute of Health, Berlin, Germany

* d.koehler@uke.de

## Abstract

### Objectives

The aim of this study was to investigate possible patterns of demand for chest imaging during the first wave of the SARS-CoV-2 pandemic and derive a decision aid for the allocation of resources in future pandemic challenges.

### Materials and methods

Time data of requests for patients with suspected or confirmed coronavirus disease 2019 (COVID-19) lung disease were analyzed between February 27th and May 27th 2020. A multinomial logistic regression model was used to evaluate differences in the number of requests between 3 time intervals (I1: 6am - 2pm, I2: 2pm - 10pm, I3: 10pm - 6am). A cosinor model was applied to investigate the demand per hour. Requests per day were compared to the number of regional COVID-19 cases.

### Results

551 COVID-19 related chest imagings (32.8% outpatients, 67.2% in-patients) of 243 patients were conducted (33.3% female, 66.7% male, mean age 60 ± 17 years). Most exams for outpatients were required during I2 (I1 vs. I2: odds ratio (OR) = 0.73, 95% confidence interval (CI) 0.62–0.86, p = 0.01; I2 vs. I3: OR = 1.24, 95% CI 1.04–1.48, p = 0.03) with an acrophase at 7:29 pm. Requests for in-patients decreased from I1 to I3 (I1 vs. I2: OR = 1.24, 95% CI 1.09–1.41, p = 0.01; I2 vs. I3: OR = 1.16, 95% CI 1.05–1.28, p = 0.01) with an acrophase at 12:51 pm. The number of requests per day for outpatients developed similarly to regional cases while demand for in-patients increased later and persisted longer.

first wave of the SARS-CoV-2 pandemic are within the paper and its Supporting Information files. As advised by our legal department, patient age and sex are not included in the supplementary data set to avoid patient identification in coherence with the given highly exact time data of our investigation at a single center. Although unlikely, identification of a single individual is deemed possible due to the low case numbers at the beginning and the end of the observation period. These data were, therefore, not submitted to protect our patients' privacy and personal rights. If this information may be beneficial to other researchers, we will gladly present the request to our institutional ethics committee and legal department. Requests may be sent to the following institutional address: Prof. Dr. med. Jin Yamamura Professor of Radiology/Co-Chair/Quality Management University Medical Center Hamburg-Eppendorf (UKE) Center for Radiology & Endoscopy Department of Diagnostic & Interventional Radiology & Nuclear medicine Martinistraße 52 20251 Hamburg j. yamamura@uke.de.

**Funding:** The authors received no specific funding for this work.

**Competing interests:** The authors have declared that no competing interests exist.

## Conclusions

The demand for COVID-19 related chest imaging displayed distinct distribution patterns depending on the sector of patient care and point of time during the SARS-CoV-2 pandemic. These patterns should be considered in the allocation of resources in future pandemic challenges with similar disease characteristics.

## Introduction

With over 60,000,000 cases and over 1,400,000 deaths globally [1] the severe acute respiratory syndrome coronavirus 2 (SARS-CoV-2) pandemic is stressing modern health care systems worldwide in an unprecedented manner. SARS-CoV-2 is mainly transmitted from person to person via droplets and direct contact [2, 3]. Real-time reverse transcription polymerase chain reaction of nasal and/or pharyngeal swabs are frequently used to make the diagnosis of a coronavirus disease 2019 (COVID-19) in patients with or without typical clinical features [4]. While many infections remain asymptomatic, COVID-19 may present with a variety of symptoms in which respiratory illness is of central importance [5, 6]. Conventional radiological chest imaging (x-ray) or computed tomography (CT) is used for disease management and triage of suspected as well as confirmed COVID-19 patients depending on symptoms and risk of disease progression [7]. Therefore, imaging plays an important role in the majority of hospitalized COVID-19 cases leading to frequent contacts of medical personnel of a radiology department with infectious patients. To prevent disease transmission, general guidance documents for the management of suspected and confirmed cases have been implemented [8, 9]. The additional workload due to the recommended safety measures and the repetitive exposure to potentially infectious patients has a negative influence on mental health of staff in the health care sector [10, 11]. While emergency department visits have declined during the early stages of the ongoing pandemic [12–14], a second wave may put more strain on the health care sector and subsequently on radiology departments. To prevent overcrowding and to preserve the wellbeing of health care professionals, efficiency in the diagnostic process of COVID-19 patients should be as high as possible.

In this explorative study, the temporal distribution of demand for chest imaging of patients with suspected or confirmed COVID-19 is analyzed as a measure for the associated increased workload in the care of potentially infectious cases in a radiology department. The derived distribution patterns could improve the allocation of resources in future pandemic challenges with a similar need for imaging.

## Materials and methods

### Patient cohort

The local institutional review board ("Ethik-Kommission der Ärztekammer Hamburg") approved this retrospective single-center study and waived the requirement for informed consent.

A total of 10522 examinations of the chest, including CT (Ingenuity Core 128, Philips Medical Systems, Cleveland, OH, USA; Somatom Force, Siemens Healthcare, Erlangen, Germany) and x-ray (GM85, Samsung Electronics, Yateley, UK; Digital Diagnost 4.1, Philips Medical Systems, Hamburg, Germany), between February 27th 2020 (first SARS-CoV-2 positive patient at this hospital) and May 27th 2020 were reviewed for requests regarding patients with

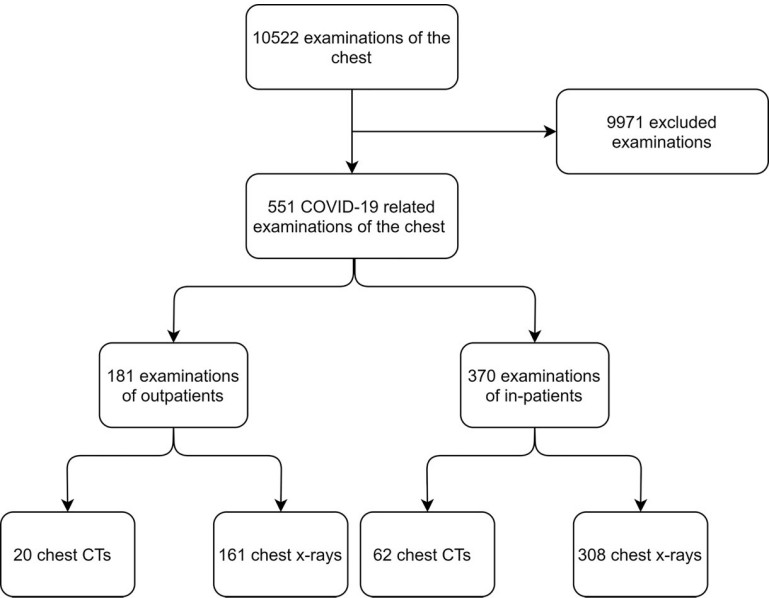

**Fig 1. Case selection flow chart.** All chest examinations of patients with suspected or confirmed coronavirus disease 2019 (COVID-19) causing additional workload due to hygienic measures were included.

suspected or confirmed COVID-19. Cases were included if a patient was either SARS-CoV-2 positive or additional hygienic measures had to be met due to the suspicion of a SARS-CoV-2 infection as stated in the electronic patient file in the hospital information system (Soarian Clinicals, Cerner Corp., Kansas City, MO, USA). The sector of patient care was defined by the requesting unit: emergency department (outpatients), general wards (in-patients), and intensive care units (in-patients). Examinations for the initial assessment of cases were differentiated from follow-up exams. A case selection flow chart is provided in Fig 1.

## Acquisition of time data

Every patient-related documentation in the radiological information system (Centricity, GE Healthcare, Chicago, IL, USA) was provided with a timestamp representing the time of creation to the minute. To adequately monitor the demand for imaging, the times of request of the included exams were collected. Requests for not-COVID-19 related chest imaging placed during the study period were included as a negative control group (n = 9562). Findings of COVID-19 related chest imaging were extracted from the written reports in the radiological information system. The case outcome evaluation was retrieved from the final report in the hospital information system.

The suspicion of COVID-19, confirmed COVID-19, or death due to COVID-19 has to be reported in correlation with the German Infection Protection Act. The resulting case data per day provided by local health authorities for the corresponding study region were retrieved from an online database of the Robert Koch Institute [15].

## Statistical analysis

Continuous data were reported using mean and standard deviation. Categorical data were summarized as absolute and relative frequencies.

Multinomial logistic regression was used to compare 3 time intervals (interval (I)1: 6 am– 2 pm, I2: 2 pm– 10 pm, I3: 10 pm– 6 am) regarding the requests for imaging per sector of patient

care (outpatients, in-patients). The odds ratio (OR) with 95% confidence interval (CI) was reported for each group comparison. A cosinor model was used to describe the diurnal distribution of requests, i.e. requests per hour of the day. This was done separately for outpatients and in-patients. To describe the goodness-of-fit, the p-value of the rhythm detection test was reported along with the correlation between observed and estimated data (r), which was used to check for the proportion of the variance explained by the rhythm. The midline statistic of rhythm (mesor or intercept), amplitude, standard error, and 95% CI as well as acrophase were reported. All p-values were descriptive due to the explorative design of the study. The statistical software R (version 3.6.1, The R Foundation for Statistical Computing) was used for all statistical analyses [16].

## Results

5.2% (n = 551) of all chest imagings during the study period were conducted because of suspected or confirmed COVID-19. Time data were available for all cases. Information concerning the study cohort comprised of 243 patients (mean age 60 ± 17 years) including 137 SARS-CoV-2 positive individuals (56.4%, mean age 59 ± 15 years) is summarized in Table 1. 82 (14.9%) of all included exams were CT scans and 469 (85.1%) x-rays. 94.2% (n = 442) of all chest radiographs were conducted using mobile x-ray units (135/30.5% outpatients, 307/69.5% in-patients). The remaining 5.8% (n = 27) were conducted using built-in radiography systems (26/96.3% outpatients, 1/3.7% in-patients). Detailed information on the included examinations is summarized in Table 2.

Most examinations for the initial assessment of suspected and confirmed COVID-19 cases were requested from the emergency department (170/84.6%). 63.6% of CT scans (n = 14) and 48% of x-rays (n = 86) for assessment showed signs of pneumonia leading to a combined positivity rate of 49.8%. 98 (48.8%) cases for initial assessment were SARS-CoV-2 positive in the final report.

The high case positivity rate of follow-up examinations (343/98%) led to a high overall positivity rate of exams of COVID-19 related cases (441/80%). 7 (2%) follow-up exams were conducted in cases that were SARS-CoV-2 negative, for example, because results of the SARS-CoV-2 testing had not been available at the time of imaging.

### Time series analysis

Demand for COVID-19 associated imaging displayed a diurnal distribution depending on the sector of patient care. Table 3 summarizes the number of requests per time interval during the study period. Most exams for outpatients in this group were required in I2 (50.8%). Less requests were observed in l1 (19.9%; I1 vs. I2: OR = 0.73, 95% CI 0.62–0.86, p = 0.01) and I3

**Table 1. Patient demographics.**

|  |  | Total (n = 243) | SARS-CoV-2 positive (n = 137) |
| --- | --- | --- | --- |
| **Sex** | Female | 81 (33.3%) | 45 (32.8%) |
|  | Male | 162 (66.7%) | 92 (67.2%) |
| **Age subgroups** | 20 – 39y | 39 (16%) | 18 (13.1%) |
|  | 40 – 59y | 74 (30.5%) | 47 (34.3%) |
|  | 60 – 79y | 101 (41.6%) | 62 (45.3%) |
|  | $\geq$ 80y | 29 (11.9%) | 10 (7.3%) |

Age subgroups reported in years (y).

**Table 2. Case data.**

| | | All examinations (n = 551) | Initial assessment (n = 201) | Follow-up (n = 350) |
|---|---|---|---|---|
| **Requesting unit** | ED | 181 (32.8%) | 170 (84.6%) | 11 (3.1%) |
| | GW | 79 (14.3%) | 19 (9.5%) | 60 (17.1%) |
| | ICU | 291 (52.8%) | 12 (6%) | 279 (79.7%) |
| **Modality** | CT | 82 (14.9%) | 22 (10.9%) | 60 (17.1%) |
| | X-ray | 469 (85.1%) | 179 (89.1%) | 290 (82.9%) |
| **Signs of pneumonia on imaging** | Positive | 428 (78%) | 100 (49.8%) | 328 (93.7%) |
| | Negative | 123 (22%) | 101 (50.2%) | 22 (6.3%) |
| **COVID-19 status** | Positive | 441 (80%) | 98 (48.8%) | 343 (98%) |
| | Negative | 110 (20%) | 103 (51.2%) | 7 (2%) |

Case data of all examinations, examinations for the initial assessment of patients with suspected or confirmed coronavirus disease 2019 (COVID-19), and follow-up examinations. Requesting unit: emergency department (ED), general wards (GW), intensive care units (ICU). COVID-19 status according to final report.

(29.3%; I2 vs. I3: OR = 1.24, 95% CI 1.04–1.48, p = 0.03). The corresponding cosinor model in Fig 2A demonstrates an acrophase at 7:29 pm (mesor 7.69 ± 0.51, 95% CI 6.7–8.69; amplitude 5.01 ± 0.73, 95% CI 3.58–6.43). Requests for COVID-19 related chest imaging of in-patients were primarily placed during I1 (52.7%) with a decreasing tendency towards I3 (I1 vs. I2: OR = 1.24, 95% CI 1.09–1.41, p = 0.01; I1 vs. I3: OR = 1.44, 95%CI 1.29–1.60, p = 0.001; I2 vs. I3: OR = 1.16, 95% CI 1.05–1.28, p = 0.01). The acrophase for in-patients was at 12:51 pm (Fig 2B; mesor 15.42 ± 1.03, 95% CI 13.39–17.44; amplitude 11.7 ± 1.46, 95% CI 8.84–14.57). The cosinor models of demand for COVID-19 related chest imaging of outpatients and in-patients demonstrated a good fit (p < 0.001 rhythm detection test) with a good percentage of explained variance (r = 0.839 and 0.868, respectively).

The acrophase for not-COVID-19 related chest imaging of outpatients was at 4:16 pm (mesor 59.96 ± 2.13, 95% CI 55.78–64.14, amplitude 41.62 ± 3.02, 95% CI 35.71–47.53; S1a Fig) and for in-patients at 1:11 pm (mesor 338.5 ± 24, 95% CI 291.4–385.5, amplitude 295.9 ± 33.94, 95% CI 229.3–362.4; S1b Fig). Both cosinor models had a good fit (p < 0.001 rhythm detection test) with a good percentage of explained variance (r = 0.949 and 0.885, respectively).

On February 28[th], the first x-ray for a suspected COVID-19 patient was requested. Figs 3 and 4 show time series of the relative number of requests for chest imaging of outpatients (n = 181) and in-patients (n = 370) with suspected/confirmed SARS-CoV-2 infection in conjunction with numbers of regional COVID-19 cases in the study region (n = 5053). Similar to

**Table 3. Temporal distribution of requests for COVID-19 related chest imaging and results of the multinomial logistic regression model.**

| Sector of patient care | 6 am—2 pm (I1) | | 2 pm—10 pm (I2) | | 10 pm—6 am (I3) | | Difference analysis | | |
|---|---|---|---|---|---|---|---|---|---|
| | Total | Percentage (95% CI) | Total | Percentage (95% CI) | Total | Percentage (95% CI) | Comparison | OR (95% CI) | p-value |
| Outpatients | 36 | 19.9% (11.7–28.1) | 92 | 50.8% (40.5–61.2) | 53 | 29.3% (19.9–38.7) | I1 vs. I2 | 0.73 (0.62–0.86) | 0.01 |
| | | | | | | | I1 vs. I3 | 0.91 (0.79–1.05) | 0.14 |
| | | | | | | | I2 vs. I3 | 1.24 (1.04–1.48) | 0.03 |
| In-patients | 195 | 52.7% (45.5–59.9) | 115 | 31.1% (24.4–37.8) | 60 | 16.2% (10.9–21.5) | I1 vs. I2 | 1.24 (1.09–1.41) | 0.01 |
| | | | | | | | I1 vs. I3 | 1.44 (1.29–1.60) | 0.001 |
| | | | | | | | I2 vs. I3 | 1.16 (1.05–1.28) | 0.01 |

Absolute and relative number of requests per time interval (I1, I2, I3) and sector of patient care including the 95% confidence interval (CI) in parentheses. Results of group comparisons between the time intervals in each sector are reported using odds ratio (OR), 95% CI in parentheses, and p-value.

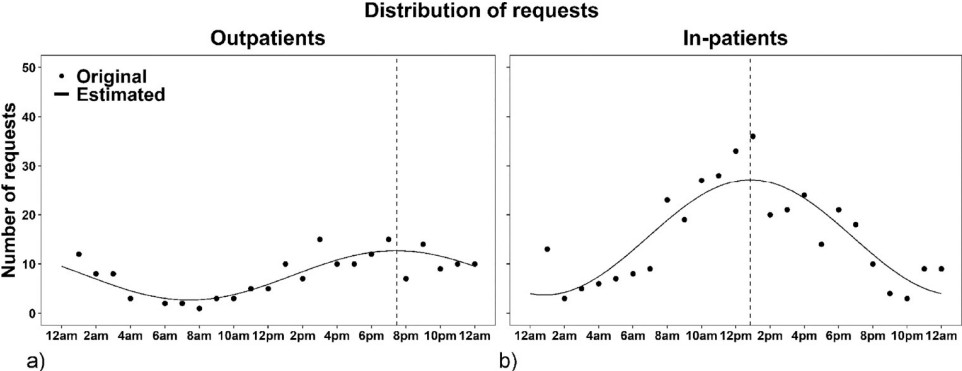

**Fig 2. Cosinor models of the demand for COVID-19 related chest imaging.** Number of requests for chest imaging of outpatients (a) and in-patients (b) with suspected or confirmed coronavirus disease 2019 (COVID-19) per hour of the day. The dotted lines indicate the acrophases at 7:29 pm (a) and 12:51 pm (b).

the development of regional cases, requests for chest imaging of suspected or confirmed COVID-19 outpatients displayed a steep increase during the second half of March. Compared to these findings, demand for imaging of in-patients demonstrated a delayed and slower increase. While case reports from the Robert Koch Institute and demand for COVID-19 related imaging of outpatients decreased steadily from the beginning of April to the end of May, requests for in-patients progressed further and persisted until the end of the study period.

Requests for chest imaging of outpatients and in-patients without an association to SARS-CoV-2 demonstrated a slight decrease throughout the time of high regional COVID-19 case numbers as depicted in S2 Fig and S3 Fig. Afterwards, the demand resumed to the level seen in the beginning of the study period.

## Discussion

In this explorative study, 551 radiological chest examinations of 243 patients with suspected or confirmed COVID-19 were analyzed in a 3-month period during the first wave of the

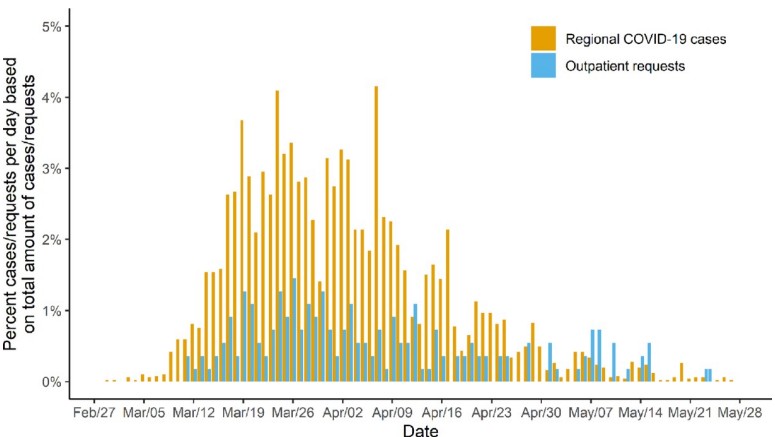

**Fig 3. Relative demand for COVID-19 related chest imaging of outpatients and relative number of regional COVID-19 cases.** Relative demand for chest imaging of outpatients with suspected or confirmed coronavirus disease 2019 (COVID-19) per day and the relative number of confirmed regional COVID-19 cases per day displayed as the percentage of the total number of requests for imaging (n = 551) and reported cases (n = 5053) between February 27th 2020 and May 27th 2020.

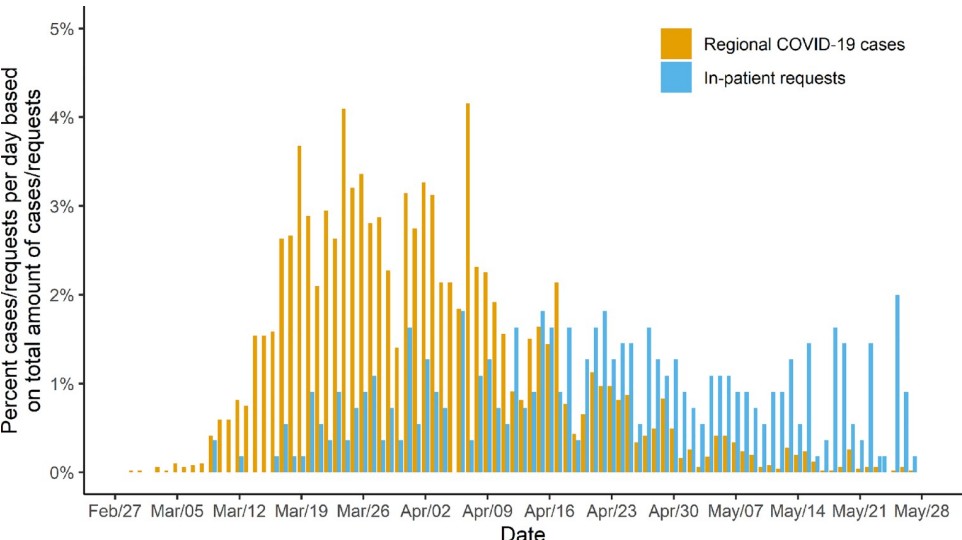

**Fig 4. Relative demand for COVID-19 related chest imaging of in-patients and relative number of regional COVID-19 cases.** Relative demand for chest imaging of in-patients with suspected or confirmed coronavirus disease 2019 (COVID-19) per day and the relative number of confirmed regional COVID-19 cases per day displayed as the percentage of the total number of requests for imaging (n = 551) and reported cases (n = 5053) between February 27th 2020 and May 27th 2020.

SARS-CoV-2 pandemic. This time series demonstrates a distinct temporal distribution of demand for chest imaging in the cohort depending on the sector of patient care and the point of time during the pandemic. COVID-19 related requests for outpatients were placed later during the day and displayed a similar development to regional cases over time. Demand for examinations of in-patients mostly adhered to working hours and showed a prolonged course after regional COVID-19 case numbers had already fallen.

The knowledge of these distribution patterns could help to manage radiology departments during a second wave or in future challenges with a similar need for imaging. Chest imaging is an essential part of the disease evaluation of COVID-19 patients [4, 5, 7]. Therefore, radiology departments are at the center of patient care during the ongoing pandemic. Due to their frequent direct patient contact, especially radiological technologists are at risk of infection and disease spread. More than 50% of the included patients in our study cohort were SARS-CoV-2 positive and frequent follow-up examinations led to multiple contacts of radiological personnel to infectious individuals (n = 441). This constitutes the importance to follow hygienic standards implemented by the Centers for Disease Control [8, 9] and consider radiologically focused recommendations [17–19]. In an effort to reduce disease transmission along transport routes, mobile x-ray units were utilized in 94.2% of all COVID-19 related chest radiographs. Next to hygienic measures, this method increases the workload of radiological technologists even more. During times of regular demand, the additional effort was compensated, but the potential work overhead during an uncontrollably spreading pandemic was already recognizable. Knoll et al. [20] have demonstrated an inverse relationship between workload and hand hygiene. Considering the complex handling of personal protective equipment in COVID-19 patients, a lower compliance with rising workload deems possible. To prevent physical and mental overload, radiology departments need to distribute personnel optimally in times of increasing case numbers. Inadequate staffing during shifts could not only have a negative influence on the adherence to hygienic standards but may also lead to a diagnostic bottleneck with a decreased turn-over in other parts of a hospital. Especially emergency departments are

prone to overcrowding, which is associated with a reduced treatment quality [21–23]. Further on, the risk of SARS-CoV-2 transmission is thought to increase with proximity to infected individuals [3, 24], and transmission events are often associated with indoor settings [25]. We hypothesize that the optimization of staffing models and workflows in radiology departments could prevent clustering and hence lower the risk of disease transmission. Results of time series analysis of the distribution of requests during a pandemic scenario may help to anticipate future patterns of demand.

In our cohort, approximately 80% of COVID-19 related chest imaging requests for outpatients occurred between 2:00 pm and 6:00 am with a peak at 7:29 pm, while demand for COVID-19 related chest imaging of in-patients mostly adhered to regular working hours (peak 12:51 pm). Only 16.2% of requests from general wards and intensive care units lay between 10:00 pm and 6:00 am. In the negative control group, demand for chest imaging of outpatients demonstrated a peak at 4:16 pm and for in-patients at 1:11 pm. Influence factors for the later presentation of COVID-19 related outpatients can only be assumed. But in our investigation of cases from the emergency department, we frequently encountered patients who already had been diagnosed with COVID-19 from physicians in private practices or outpatient clinics delaying the presentation at our institution, even compared to other outpatients. The similar peaks of requests for imaging of in-patients, independent of their COVID-19 status, indicate that constant variables (e.g. staffing models) may be more important influencing factors than a certain diagnosis in this group.

Depending on the indication and the sector of patient care, the number of requests for chest imaging per day differed substantially. COVID-19 related demand for imaging of outpatients developed similarly to case reports of infections in the study region, whereas requests for in-patients displayed a later and slower increase with a persistent demand until the end of the study period. On the other hand, requests for chest imaging of in- as well as outpatients without an association to SARS-CoV-2 demonstrated a slight decrease during times of high regional COVID-19 case numbers. Possible causes for this development may be the decline of emergency department visits as reported in various publications [12–14] and the reduction of capacities for elective procedures.

Considering these findings may be beneficial in future pandemic scenarios with a similar need for imaging. Personnel should be redistributed from early to late/night shifts during the first phase of a pandemic to meet the predominantly late demand for imaging of outpatients. Once regional case numbers decrease, staffing models could be readjusted to regular working hours to process the increasing number of requests for imaging of in-patients. These changes may help to reduce the workload per employee, to prevent disease transmission, and to save resources in the long run while preserving a high quality of patient care.

The retrospective nature of our study has some limitations. The assumed relationship between case numbers of a region and requests for imaging in a department underly multiple influencing factors and may be biased, for example, due to selection. Our data represent the information of one institution with a limited number of patients during a controllable period of the SARS-CoV-2 pandemic. Dynamics may vary at other centers, with higher case numbers, or different diseases. Nevertheless, the presented cohort generated a sufficient number of time data points (n = 551) to demonstrate differences between the subgroups, displaying the potential of time series analysis in the investigated setting. To evaluate the benefit of the proposed approach, however, prospective investigations at multiple centers and interventions (e.g. changes to staffing models) will be necessary.

## Conclusions

The demand for COVID-19 related chest imaging during the first wave of the SARS-CoV-2 pandemic showed a characteristic temporal distribution. Imaging requests for outpatients were usually placed later during the day compared to in-patients. Moreover, daily requests for outpatients developed similarly to regional cases over time. Compared to this, the demand for imaging of in-patients increased later but persisted when regional case numbers had already fallen. These patterns should be considered in the creation of staffing models in future pandemic challenges with a similar need for radiological imaging. Adjustments to the distribution of resources may help to maintain high standards of patient care and ensure staff safety.

## Supporting information

**S1 Table. Demand for COVID-19 related chest imaging and regional COVID-19 case numbers.** Number of requests for coronavirus disease 2019 (COVID-19) related chest imagings per day (total, outpatients, in-patients) and number of regional COVID-19 cases per day provided by the Robert Koch Institute [15].
(DOCX)

**S1 Fig. Cosinor models of the demand for not-COVID-19 related chest imaging.** Number of requests for chest imaging of outpatients (a) and in-patients (b) without suspected or confirmed coronavirus disease 2019 (COVID-19) per hour of the day. The dotted lines indicate the acrophases at 4:16 pm (a) and 1:11 pm (b).
(TIF)

**S2 Fig. Relative demand for not-COVID-19 related chest imaging of outpatients and relative number of regional COVID-19 cases.** Relative demand for chest imaging of outpatients without suspected or confirmed coronavirus disease 2019 (COVID-19) per day and the relative number of confirmed regional COVID-19 cases per day displayed as the percentage of the total number of requests for imaging (n = 1439) and reported cases (n = 5053) between February 27th 2020 and May 27th 2020.
(TIF)

**S3 Fig. Relative demand for not-COVID-19 related chest imaging of in-patients and relative number of regional COVID-19 cases.** Relative demand for chest imaging of in-patients without suspected or confirmed coronavirus disease 2019 (COVID-19) per day and the relative number of confirmed regional COVID-19 cases per day displayed as the percentage of the total number of requests for imaging (n = 8123) and reported cases (n = 5053) between February 27th 2020 and May 27th 2020.
(TIF)

**S1 File. Time data of requests for COVID-19 related chest imaging during the study period.**
(XLSX)

**S2 File. Time data of requests for not-COVID-19 related chest imaging during the study period.**
(XLSX)

## Author Contributions

**Conceptualization:** Daniel Koehler, Jin Yamamura.

**Data curation:** Daniel Koehler, Isabel Molwitz, Philipp May, Hanna Maria Görich.

**Formal analysis:** Daniel Koehler, Ann-Kathrin Ozga.

**Investigation:** Daniel Koehler.

**Methodology:** Daniel Koehler.

**Project administration:** Jin Yamamura.

**Software:** Ann-Kathrin Ozga.

**Supervision:** Jin Yamamura.

**Validation:** Sarah Keller, Gerhard Adam, Jin Yamamura.

**Visualization:** Daniel Koehler.

**Writing – original draft:** Daniel Koehler.

**Writing – review & editing:** Ann-Kathrin Ozga, Isabel Molwitz, Philipp May, Hanna Maria Görich, Sarah Keller, Gerhard Adam, Jin Yamamura.

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
