## [Decision Letter · Decision Letter 0]

6 Jan 2021

PONE-D-20-38040

Time series analysis of the demand for COVID-19 related chest imaging during the first wave of the SARS-CoV-2 pandemic: an explorative study

PLOS ONE

Dear Dr. Koehler,

Thank you for submitting your manuscript to PLOS ONE. After careful consideration, we feel that it has merit but does not fully meet PLOS ONE’s publication criteria as it currently stands. Therefore, we invite you to submit a revised version of the manuscript that addresses the points raised during the review process.

We look forward to receiving your revised manuscript.

Kind regards,

Ruslan Kalendar, PhD

Academic Editor

PLOS ONE

Journal Requirements:

2.We note that you have indicated that data from this study are available upon request. PLOS only allows data to be available upon request if there are legal or ethical restrictions on sharing data publicly. For information on unacceptable data access restrictions, please see http://journals.plos.org/plosone/s/data-availability#loc-unacceptable-data-access-restrictions.

Reviewers' comments:

Reviewer's Responses to Questions

**Comments to the Author**

1. Is the manuscript technically sound, and do the data support the conclusions?

Reviewer #1: Partly

Reviewer #2: Yes

2. Has the statistical analysis been performed appropriately and rigorously? 

Reviewer #1: Yes

Reviewer #2: Yes

3. Have the authors made all data underlying the findings in their manuscript fully available?

Reviewer #1: Yes

Reviewer #2: Yes

4. Is the manuscript presented in an intelligible fashion and written in standard English?

Reviewer #1: Yes

Reviewer #2: Yes

5. Review Comments to the Author

Reviewer #1: 

The main motivation of Koehler et al., was to uncover patterns of CT imaging requests demands of confirmed or ‘very likely’ COVID +ve patients and to predict future allocation of resources during a global pandemic.

Some Major suggestions:

• The study cohort that the authors end up using for this analysis comprises of only 243 individuals. Since, 98% of all final examinations are COVID-positive, the authors could be more inclusive and perhaps relax the criteria for filtering out non-COVID related CT imaging requests. That could perhaps increase the study cohort size and provide more robust patterns.

• Second, it is also worth comparing the CT imaging demands in the negative set (i.e CT imaging requests that are not COVID-related). That would be a good negative control experiment and would rule out any inherent bias present in in-patient and out-patient CT imaging demand patterns.

Some Minor comments

• The table-3 was chopped off the pdf that was given to the reviewers. Perhaps, the authors can split it out into multiple tables (might help the readers).

• The authors claim that their analysis will help “aid for the allocation of resources in future pandemic challenges”. However, not all “future” pandemics would have pattern correlated with CT imaging. The authors should perhaps tone down the score of their study.

Reviewer #2: 

This is a well written paper. The work is straightforward but has important implications. I could hardly find anything to be critical of. My only concern would be the limited number of data available. Perhaps more datasets can be included now.

---

## [Author Response · Author response to Decision Letter 0]

9 Feb 2021

Dear Dr. Kalendar,

Dear Reviewers,

thank you very much for considering our manuscript and for the helpful reviews of our work. We tried to address each point and are very pleased to submit our revised draft, which has been substantially improved as a result of your reviews.

Response to the Editor 

If there are ethical or legal restrictions on sharing a de-identified data set, please explain them in detail (e.g., data contain potentially identifying or sensitive patient information) and who has imposed them (e.g., an ethics committee). Please also provide contact information for a data access committee, ethics committee, or other institutional body to which data requests may be sent.

All data regarding the temporal distribution of demand for COVID-19 related chest imaging at our institution during the first wave of the SARS-CoV-2 pandemic were fully submitted. All information needed to reproduce our statistical analyses or to investigate this matter differently are shared without restriction. This includes:

3) The points extracted from images for analysis.

As advised by our legal department, patient age and sex are not included in the supplementary data set to avoid patient identification in coherence with the given highly exact time data of our investigation at a single center. Although unlikely, identification of a single individual is deemed possible due to the low case numbers at the beginning and the end of the observation period. These data were, therefore, not submitted to protect our patients’ privacy and personal rights. Nevertheless, we see the obligation to publish the results of our analysis of the distribution of age and sex in our collective to support the body of evidence regarding COVID-19 in general. If this information may be beneficial to other researchers, we will gladly present the request to our institutional ethics committee and legal department. Requests may be sent to the following institutional address:

Prof. Dr. med. Jin Yamamura 

Professor of Radiology/Co-Chair/Quality Management

University Medical Center Hamburg-Eppendorf (UKE)

Center for Radiology & Endoscopy

Department of Diagnostic & Interventional Radiology & Nuclear medicine

Martinistraße 52

20251 Hamburg

j.yamamura@uke.de

Responses to Reviewer #1 

• The study cohort that the authors end up using for this analysis comprises of only 243 individuals. Since, 98% of all final examinations are COVID-positive, the authors could be more inclusive and perhaps relax the criteria for filtering out non-COVID related CT imaging requests. That could perhaps increase the study cohort size and provide more robust patterns.

We agree that the size of our study cohort from a single center during the fortunately still controllable part of the SARS-CoV-2 pandemic is a limitation of our investigation. Nonetheless, our institution was confronted with never-before-seen numbers of potentially infectious patients. To identify times of increased workload due to hygienic measures, we chose strict inclusion criteria for our study cohort. Every patient file was thoroughly searched for a patient’s infection status at the time of presentation at the department of radiology. We fear that relaxing these criteria might decrease the accuracy of the presented distribution patterns and reduce the validity of our conclusions. We would like to highlight that requests for the included 243 individuals exceeded the absolute number of patients substantially (n = 551). Differences between the investigated time intervals were seen with one exception (outpatients: time interval I1 vs. I3, p = 0.14). The derived cosinor models of the temporal distribution of requests depicted a good fit (p < 0.001) with a good percentage of explained variance (r = 0.839 and 0.868, respectively). These results indicate that the number of the included patients produced a sufficient amount of data to demonstrate differences between the investigated groups and time intervals. 

If a larger study cohort would be deemed necessary to support our findings, we would like to propose an expansion of the study period until September 2020 (begin of the second wave). Although numbers of newly diagnosed regional COVID-19 cases decreased as depicted in Figs 3 and 4 of our manuscript, this would be the most adequate way to increase case numbers. If this approach would be applicable, we would like to ask for more time to complete the review of our manuscript. 

• Second, it is also worth comparing the CT imaging demands in the negative set (i.e CT imaging requests that are not COVID-related). That would be a good negative control experiment and would rule out any inherent bias present in in-patient and out-patient CT imaging demand patterns.

Thank you very much for this very helpful suggestion. We analyzed the not-COVID-19 related requests for chest imaging during the study period and differentiated between outpatients and in-patients. Interestingly, the resulting cosinor models demonstrated differences between the temporal distribution of demand for COVID-19 related and not-COVID-19 related requests for outpatients (acrophases at 7:29 pm and 4:16 pm, respectively). A possible explanation for this delay may be the additional time needed to perform SARS-CoV-2 tests before imaging was requested. Demand for in-patients did not differ substantially (acrophases at 12:51 pm and 1:11 pm, respectively) indicating that constant influencing factors (e.g. staffing models) may be more important than a certain diagnosis. Requests for chest imaging in the negative control group showed a slight inverse relationship with regional COVID-19 case numbers both for outpatients and in-patients. Testing for statistical significance with a multinomial logistic regression model was not included to preserve the focus of the original investigation and avoid the problem of multiple testing.

However, the negative control supports our conclusion that the temporal distribution of demand for COVID-19 related imaging follows a pattern depending on the sector of patient care and the time during the pandemic. Regional case data may be seen as a criterion to adjust the distribution of resources in times of a pandemic with a similar need for imaging. 

• The table-3 was chopped off the pdf that was given to the reviewers. Perhaps, the authors can split it out into multiple tables (might help the readers).

Table 3 was refitted and should be visible in full extend in the revised manuscript.

• The authors claim that their analysis will help “aid for the allocation of resources in future pandemic challeng-es”. However, not all “future” pandemics would have pattern correlated with CT imaging. The authors should perhaps tone down the score of their study.

Thank you very much for this important feedback. The central position of radiology departments in the current pandemic may not be assumed in all future scenarios. The statement was adjusted accordingly. 

Response to Reviewer #2 

This is a well written paper. The work is straightforward but has important implications. I could hardly find anything to be critical of. My only concern would be the limited number of data available. Perhaps more datasets can be included now.

Thank you very much for this positive review of our manuscript. We also see the size of our study cohort as a limitation of our work. However, our analysis primarily seeks to investigate time data. Due to repetitive imaging of COVID-19 patients, we were able to accumulate a total of 551 time points. The resulting analyses showed significant differences between the investigated time intervals with only one exception (outpatients: time interval I1 vs. I3, p = 0.14). The cosinor models of demand demonstrated a good fit (p < 0.001) with good values of explained variance (r = 0.839 and 0.868, respectively). These results indicate that the number of data points derived from our study cohort is sufficient to show differences between the investigated groups and time intervals.

If a larger study cohort would be deemed necessary to support our findings, we would like to propose to expand the study period until September 2020 after which the second wave of the SARS-CoV-2 pandemic began. Although numbers of newly diagnosed regional COVID-19 cases decreased as depicted in Figs 3 and 4 of our manuscript, this would be the most adequate way to increase case numbers. If this approach would be applicable, we would like to ask for more time to complete the review of our manuscript. 

On behalf of all co-authors and myself, I thank you very much for your profound reviews and hope you will find our revised manuscript acceptable for publication.

Kind regards,

Daniel Koehler

---

## [Editor Report · Decision Letter 1]

11 Feb 2021

Time series analysis of the demand for COVID-19 related chest imaging during the first wave of the SARS-CoV-2 pandemic: an explorative study

PONE-D-20-38040R1

Dear Dr. Koehler,

We’re pleased to inform you that your manuscript has been judged scientifically suitable for publication and will be formally accepted for publication once it meets all outstanding technical requirements.

Kind regards,

Ruslan Kalendar, PhD

Academic Editor

PLOS ONE

---

## [Editor Report · Acceptance letter]

17 Feb 2021

PONE-D-20-38040R1 

Time series analysis of the demand for COVID-19 related chest imaging during the first wave of the SARS-CoV-2 pandemic: an explorative study 

Dear Dr. Koehler:

I'm pleased to inform you that your manuscript has been deemed suitable for publication in PLOS ONE. Congratulations! Your manuscript is now with our production department. 

Kind regards, 

on behalf of

Prof. Ruslan Kalendar 

Academic Editor

PLOS ONE